# Activity of Propolis Nanoparticles against HSV-2: Promising Approach to Inhibiting Infection and Replication

**DOI:** 10.3390/molecules27082560

**Published:** 2022-04-15

**Authors:** Sirikwan Sangboonruang, Natthawat Semakul, Sanonthinee Sookkree, Jiraporn Kantapan, Nicole Ngo-Giang-Huong, Woottichai Khamduang, Natedao Kongyai, Khajornsak Tragoolpua

**Affiliations:** 1Division of Clinical Microbiology, Department of Medical Technology, Faculty of Associated Medical Sciences, Chiang Mai University, Chiang Mai 50200, Thailand; sirikwan.sang@cmu.ac.th (S.S.); sanonthinee_sookkree@cmu.ac.th (S.S.); woottichai.k@cmu.ac.th (W.K.); 2Infectious Diseases Research Unit (IDRU), Faculty of Associated Medical Sciences, Chiang Mai University, Chiang Mai 50200, Thailand; 3Department of Chemistry, Faculty of Science, Chiang Mai University, Chiang Mai 50200, Thailand; natthawat.semakul@cmu.ac.th; 4Molecular Imaging and Therapy Research Unit, Department of Radiologic Technology, Faculty of Associated Medical Sciences, Chiang Mai University, Chiang Mai 50200, Thailand; jiraporn.kan@cmu.ac.th; 5Associated Medical Sciences (AMS)-CMU IRD Research Collaboration, Chiang Mai 50200, Thailand; nicole.ngo-giang-huong@phpt.org; 6Maladies Infectieuses et Vecteurs: Écologie, Génétique, Évolution et Contrôle (MIVEGEC), University of Montpellier, Centre National de la Recherche Scientifique (CNRS), Institut de Recherche Pour le Développement (IRD), 34394 Montpellier, France

**Keywords:** herpes simplex virus type 2, antiviral activity, propolis, chitosan, poly(lactic-co-glycolic acid), polymeric nanoparticles

## Abstract

Herpes simplex type 2 (HSV-2) infection causes a significant life-long disease. Long-term side effects of antiviral drugs can lead to the emergence of drug resistance. Thus, propolis, a natural product derived from beehives, has been proposed to prevent or treat HSV-2 infections. Unfortunately, therapeutic applications of propolis are still limited due its poor solubility. To overcome this, a nanoparticle-based drug delivery system was employed. An ethanolic extract of propolis (EEP) was encapsulated in nanoparticles composed of poly(lactic-co-glycolic acid) and chitosan using a modified oil-in-water single emulsion by using the solvent evaporation method. The produced nanoparticles (EEP-NPs) had a spherical shape with a size of ~450 nm and presented satisfactory physicochemical properties, including positively charged surface (38.05 ± 7.65 mV), high entrapment efficiency (79.89 ± 13.92%), and sustained release profile. Moreover, EEP-NPs were less cytotoxic on Vero cells and exhibited anti-HSV-2 activity. EEP-NPs had a direct effect on the inactivation of viral particles, and also disrupted the virion entry and release from the host cells. A significant decrease in the expression levels of the HSV-2 replication-related genes (*ICP4*, *ICP27*, and *gB*) was also observed. Our study suggests that EEP-NPs provide a strong anti-HSV-2 activity and serve as a promising platform for the treatment of HSV-2 infections.

## 1. Introduction

Herpes simplex virus type 2 (HSV-2) is considered to be one of the most severe pathogens in humans causing an important sexually transmitted disease known as genital herpes. Notably, infection with HSV-2 has been declared high risk for HIV infection, as well as invasive cervical carcinoma [1]. The mechanisms related to viral replication and new virion assembly and release are important factors of HSV-2 infection. Following the entry of viral genomes into the nucleus, immediate early (IE), early (E), and late (L) genes lead to the generation of mature virions [2].

HSV-2 infections are life-long, and the current standard treatment relies on acyclovir (ACV) and related synthetic nucleoside analogs that target the viral polymerase [2]. However, these drugs cannot cure the infections; thus, reactivation can occur, particularly in immunocompromised individuals. Furthermore, long-term treatment with these drugs can lead to drug resistance [3,4]. To overcome this problem, several strategies have been proposed on other targets. Therefore, the use of natural products has gained much attention.

Natural products are a great source of several bioactive molecules and are usually used as alternative treatments due to their safety, therapeutic properties, and low cost [3,5]. Moreover, they have also proved to be a promising way to overcome challenges with drug resistance since their mode of action has been reported in various ways [1]. Propolis is a natural resinous substance produced by bees that has been widely used as a traditional medicine for centuries [6]. Numerous components have been identified in propolis, mainly polyphenol (flavonoids, phenolics, and esters), depending on the bee species, botanic and geographic origin [7,8]. Investigation of active ingredients in brown propolis was earlier performed by our group. The chromatogram profile analyzed by means of HPLC presented its chemical compositions: gallic acid, quercetin, pinocembrin, chrysin, and galangin [9]. Moreover, propolis has been shown to display potent biological properties, such as antimicrobial, anti-inflammatory, anticancer, and antioxidant properties. Notably, its antiviral effect has also been proven [5,10]. The antiviral activity of propolis has been investigated against pathogenic human viruses such as influenza [11], human immunodeficiency virus (HIV) [12], human coronaviruses (SARS-CoV-2) [5], and herpes simplex viruses (HSV-1 and HSV-2) [13]. It was suggested that propolis inhibits HSV-2 infection through the destruction of the viral envelope and cellular absorption [13,14]. However, dissolubility of propolis has limited its therapeutic application and needs to be improved [15].

In recent years, nanotechnology has been extensively applied in pharmaceutical and medical sciences by using nanoparticle-based drug delivery systems. These systems not only provide an efficient carrier, but also resolve the problem of dissolution of hydrophobic substances, resulting in an elevated level of therapeutic efficacy [16,17]. In our previous work conducted by Iadnut et al., 2019, a polymeric nanoparticle platform of a poly(lactic-co-glycolic acid) (PLGA) formulation was employed to encapsulate the ethanolic extract of propolis (EEP). This nanoformulation possessed a great characteristic of lower cytotoxicity, and the water solubility of EEP was also improved. However, a weak point associated with the zeta potential (ZP) of the nanoparticle surface was present and required a charge modification to stabilize its nanostructure [9]. Modification of the surface charge can be rendered by the addition of negative or positive charge-inducing agents [18]. In this study, chitosan (CS), a natural biopolymer, was used to produce cationic nanoparticles (NPs) that are known to have low toxicity with nonimmunogenic and biodegradable nature [19,20]. Besides an increase in the ZP, we also hypothesized that CS-modified PLGA NPs loaded with EEP could influence HSV-2 activity through the disruption of viral replication-related factors such as the IE or L genes.

Thus, this work was aimed at encapsulating EEP within CS-modified PLGA NPs (EEP-NPs) and characterizing their physicochemical properties, anti-HSV-2 activity, and mechanisms of actions.

## 2. Results

### 2.1. Physicochemical Characterization of the EEP-NPs

#### 2.1.1. Particle Size, Polydispersity Index (PDI), Zeta Potential (ZP), Entrapment Efficiency (EE), Loading Capacity (LC), and Morphology

EEP was encapsulated in a PLGA-based formulation as described in our previous work [9]. To produce a repulsive force between the particles, CS was incorporated into the formulations. As shown in Table 1 and Figure 1, the modification of the NPs with CS showed the average particle size as approximately 650 nm, with a uniform particle size distribution at the PDI value of 0.34 for the Empty-NPs, while the EEP-NPs had a particle size of 450 nm with a PDI of 0.21. Importantly, the ZP of their surface charge improved to be positive, between +36 and +38 mV. The encapsulation of EEP resulted in the EE and LC values of 80% and 40%, respectively. Moreover, the morphology of the NPs was spherical in shape and had a smooth surface (Figure 1).

#### 2.1.2. EEP Release In Vitro

The in vitro release profile of EEP from the NPs was determined during a 24 h experiment in PBS at pH 4.0 and pH 7.4. As shown in Figure 2, the release pattern of the EEP-NPs showed the maximum cumulative release of EEP after 3 h, followed by a sustained release phase up to 24 h. The sustained release reached approximately 30% with no difference in the percentage of the EEP release between different conditions of pH 7.4 and pH 4.0. This result suggests that the EEP-NPs released EEP in a sustained manner.

### 2.2. In Vitro Evaluation of Anti-HSV-2 Activity

Prior to performing the antiviral assay, the cytotoxicity of the EEP-NPs (or the Empty-NPs) was examined. In this study, Vero cells were used as the host cells for HSV-2 infection, and cell viability was determined by means of the trypan blue exclusion assay. As shown in Figure 3a, cellular cytotoxicity of the EEP-NPs and Empty-NPs was not observed in a concentration ranging from 0 to 1.25 mg/mL. However, the cell viability was significantly reduced with the concentration of the NPs at 2.5 mg/mL by 40% and 20% after treatment with the EEP-NPs and the Empty-NPs, respectively (*p* < 0.05), and the 50% cytotoxicity concentrations (CC_50_) of both NPs were more than 2.5 mg/mL (Table 2). Using this result, concentrations of the NPs lower than 2.5 mg/mL were used to further study the antiviral activity.

To investigate the anti-HSV-2 activity of the EEP-NPs, the infected cells were treated with various concentrations of the NPs for 24 h, and the reduction of plaque formation was used to evaluate the anti-HSV-2 activity by the EEP-NPs. As shown in Figure 3b and Table 2, the EEP-NPs inhibited the plaque formation and showed a significant percentage of plaque inhibition in a dose-dependent manner with the 50% inhibition concentration (IC_50_) at 0.80 ± 0.16 mg/mL. Meanwhile, the Empty-NPs had no effect. The selectivity index (SI) of the EEP-NPs was more than 3.12, indicating a nontoxic agent, and further studies can be performed. In addition, the administration of the EEP-NPs at 1.25 mg/mL showed a comparable activity to ACV (IC_90_). Thus, the highest concentration of the NPs (1.25 mg/mL) was used to investigate the mode of action in further experiments.

### 2.3. Cellular Uptake and Localization of the EEP-NPs

To visualize the cellular uptake of the nanoformulation, Nile red (NR)-labeled particles were used to treat the infected cells. Cellular uptake was found in the cells and mainly accumulated in the cell membrane, indicated by the FITC–α_v_-integrin membrane protein (Figure 4). This implied that the EEP-NPs were uptaken into the cells, and the gathering of the EEP-NPs was observed in the cell membrane and the cytoplasm, but not found in the nucleus.

### 2.4. Effect of the EEP-NPs on Different Modes of HSV-2 Infection

The direct effect of the EEP-NPs on HSV-2 particles was investigated at different timepoints of the treatment. The viral particles were incubated with the EEP-NPs (or the Empty-NPs) for 0 min, 30 min, 6 h, and 24 h, followed by virus inoculation on the cells. The plaque formation assay was performed to determine the inhibitory effect of the EEP-NPs against HSV-2. Treatment with the NPs significantly inhibited plaque formation in a time-dependent manner in each treatment period when compared to the virus control (Figure 5a). Moreover, the EEP-NPs significantly reduced the plaque formation as compared to the Empty-NPs after 6 and 24 h of exposure (*p* < 0.05). This suggests that the EEP-NPs directly inhibited HSV-2 particles.

Besides their ability to directly inactivate HSV-2, the EEP-NPs affected the virus entry and release. The Vero cells were preincubated with the EEP-NPs (or the Empty-NPs) for 24 h before applying the virus to the cells. In comparison to the virus control, the EEP-NPs and the Empty-NPs showed a remarkable plaque inhibition of approximately 60% and 80%, respectively. Moreover, the percentage of inhibition was significantly lower in the EEP-NPs when compared to the Empty-NPs (Figure 5b). This result implies that both NPs might block the entry of the virus into the cells.

To determine whether the EEP-NPs could influence the virus release, titers of extracellular and intracellular virus were measured in the presence and absence of the EEP-NPs (or the Empty-NPs). As shown in Figure 5c, the remaining intracellular virus was observed in the Empty-NPs sample, which was insignificant compared to the virus control. Nevertheless, the HSV-2-infected cells and the cells treated with the EEP-NPs demonstrated a significant number of plaques. As the rate of release was determined, the reduction rate of the Empty-NPs showed insignificant values when compared to the virus control. Meanwhile, the EEP-NPs statistically decreased the release activity when compared to the Empty-NPs (*p* < 0.05) and the virus control (*p* < 0.001) (Figure 5d). These results suggest that the EEP-NPs could block and/or inactivate intracellular and extracellular virus particles. However, the Empty-NPs could obstruct the virus particle release outside the host cells, but not inactivate intracellular virus particles. This result revealed that the EEP-NPs were able to exhibit dual functions by blocking the virus release and reducing the infectivity of HSV-2.

To confirm the ability of the EEP-NPs to block HSV-2 release, a fluorescent imaging analysis was performed. Noticeably, in both uninfected and HSV-2-infected cells, the EEP-NPs were visibly intense at the cell membrane (Figure 6), suggesting that the EEP-NPs might block virus release out to the extracellular space after infection.

### 2.5. Determination of the Virus Titer Using a Plaque Assay

As a result of the decrease in plaque formation, we hypothesized that the EEP-NPs might affect the replication of HSV-2. Thus, the molecular expression levels of HSV-2 virus-associated factors, including the *ICP4*, *ICP27* and *gB* genes, were further investigated. As shown in Figure 7, the mRNA expression of the targeted genes showed a significant reduction after treatment with the NPs, particularly the EEP-NPs, as compared to the virus control. In comparison to the Empty-NPs, the expression levels of IE genes *ICP4* and *ICP27* were statistically suppressed 0.6-fold (*p* < 0.05) and 6.6-fold (*p* < 0.001), respectively. Then, the subsequent expression of *gB,* which is an L gene, was significantly downregulated (*p* < 0.001) approximately 35-fold. These findings suggest that NPs affect the expression of HSV-2 replication-related genes. Furthermore, the inhibitory effect on these genes was also promoted due to the action of EEP released from the formulation and the diminished levels of IE genes, consequently resulting in the downregulation of the L gene.

## 3. Discussion

Natural product-based nanoparticles have been proposed as alternatives to treat infections and overcome the problem of drug resistance [21,22,23]. We developed a nanoparticle-based drug delivery system encapsulating propolis and analyzed the anti-HSV-2 properties of these nanoparticles.

Propolis serves as a source of numerous active molecules with broad-spectrum antimicrobial activity, as well as antiviral properties. The effect of propolis against HSV-2 has been reported in many studies. It was demonstrated that propolis extracts from the bee glue of *Apis mellifera* and Canadian propolis have a virucidal effect and interrupt HSV-2 activity in the virus adsorption step [14,24]. Yildirim et al., 2016, also observed the suppression of HSV-2 replication by Hatay propolis along with the combination with ACV, showing a synergetic antiviral activity that was superior to the treatment with ACV alone [25]. Although propolis has proved its powerful activity, its poor solubilization has been noted [15].

We previously showed that EEP can be successfully loaded within PLGA NPs to improve its water solubility; however, these nanoparticles were unstable due to the insufficient ZP value of the surface charge [9]. The surface charge depends on the type of constituents used in the NPs. CS, a cationic biopolymer, is widely used to modify PLGA-based NPs via ionic crosslinking between the positive amino groups present in the CS chains and an oppositely charged agent [26,27]. Based on the previous formulation of PLGA NPs [9], CS was added into the formulation to create a ZP of the NP surface. The modified formulation was improved as to be positively charged with the ZP value of approximately +38 mV resulting from the characteristics of CS and properly presented the average size at 450 nm and a PDI value of 0.21. In addition, these nanoparticles were able to release EEP in a sustained manner at pH 7.4 (stimulating the pH of a normal human body) and pH 4.0 (representing a vaginal pathological condition). In concordance with a previous study [9], rapid release was observed within the first 3 h which lasted for up to 24 h. Nonetheless, the release rate was lower than what was reported. This difference in the release rate may have been due to the adsorption of drugs on CS through charge interaction which may influence the reduction of drug leakage [27,28].

Since the biological study was performed, the cytotoxicity of the EEP-NPs was first examined. In concordance with the previous work, the EEP-NPs presented less toxicity to the Vero cells [9]. These results indicated a suitable biomedical application of PLGA and CS as nanocarriers because of their nontoxic, biodegradable, and biocompatibility properties [27]. Moreover, the safety of EEP was widely reported as well as in Vero cells [29,30].

Subsequently, low toxic concentrations of the EEP-NPs were tested for antiviral activity. Our results showed the antiviral effects of the EEP-NPs after treatment of the HSV-2-infected cells for 24 h in a dose-dependent manner. Moreover, the SI value also indicated nontoxicity of the EEP-NPs as previously described [31]. However, many variations of acceptance criteria of SI values have been reported. Thus, further investigation is still required. Then, the CLSM analysis was conducted to show the uptake of the NPs into the infected cells (Figure 4). The result demonstrated that the EEP-NPs could enter the infected cells and were located mainly in the cell membrane, followed by the cytoplasm.

The role of the EEP-NPs against HSV-2 was further investigated by different modes of infection. The restriction of viral particles by the EEP-NPs was first elucidated. The significant direct inactivation effect on free HSV-2 by the EEP-NPs occurred within the first 30 min and up to 24 h of exposure. The Empty-NPs also inactivated HSV-2 virions, possibly due to the degradation of the NPs and the release of cationic CS.

Likewise, after the preincubation of the cells with the cationic NPs (EEP-NPs or Empty-NPs) with a negatively charged cell surface [27], ultimately, the viral attachment, adsorption, and entry processes were blocked. Noticeably, the Empty-NPs showed a higher activity than the treatment with the EEP-NPs in this step. This effect was probably influenced by the dissociation of CS from the Empty-NPs, leading to ionic adsorption between CS and the cell surface, while incorporation of EEP in the formulation might result in the interference with or neutralization of the surface charge of CS; therefore, accumulation of CS on the cell surface may be reduced. Moreover, in the viral release assay, some intracellular and extracellular viruses were retained after treatment with the Empty-NPs. Noticeably, the EEP-NPs eliminated almost all of the intracellular and extracellular viruses as shown in Figure 5c. These results indicated that both the EEP-NPs and the Empty-NPs disrupted the viral release affecting budding, or cell-to-cell spread of viruses. These assumptions were consistent with the intense fluorescence of the EEP-NPs mostly at the membrane (Figure 6). This effect can be explained by the ionic adsorption between the characteristics of positively charged CS-based NPs and negatively charged cell membranes [27,32].

Our findings can be explained by two main aspects. Firstly, CS has an intrinsic antimicrobial property including antiviral activity [23]. It has been proposed that the electrostatic interaction of positively charged CS and the negatively charged surface of the virus can inhibit the viral activity and/or directly kill the virus through viral protective membrane disruption [23,33]. Moreover, the viral capsid proteins or other virus-specific proteins could be interrupted by CS or its derivatives, leading to the prevention of viral glycoproteins and the interaction with their receptors, the reduction of viral entry, and eventually the suppression of viral replication [23,34,35]. Secondly, the active constituents of EEP released from nanocarriers might directly affect the virion envelope, capsid proteins, or mask viral compounds such as glycoproteins, which are responsible for adsorption or entry into host cells, as described elsewhere [14,36]. It was stated that the constituents of propolis, galangin, and chrysin are responsible for antiviral activity by reducing the plaque formation of free HSV. However, the effect of propolis on virus replication was not detected [37]. Thus, we hypothesized whether the EEP-NPs were involved in HSV-2 replication.

After entering the host cells, the viral genome is transported and released into the nucleus where the replication cycle takes place. HSV replication cascade is a sequential expression of the IE, E, and L gene products. Initially, the *IE* genes are present (2–4 h of infection), and the IE proteins are required to trigger the transcription of the *E* (4–9 h of infection) and *L* genes (24 h of infection), respectively. Among the five viral IE regulatory proteins (ICP0, ICP4, ICP22, ICP27, and ICP47), ICP4 and ICP27 are phosphoproteins that are required to induce the expression of the *E* and *L* genes and are essential for HSV replication and infection [38,39,40]. Moreover, late-stage protein gB was also chosen to be analyzed according to the low activity of viral release. Late-stage protein gB, a class III fusion glycoprotein, acts as a viral fusogen. The fusion reaction delivers the viral nucleocapsid and tegument proteins into the host cell [41]. As shown in the result, after the administration of the EEP-NPs (or the Empty-NPs), the mRNA expression levels of the related factors, *ICP4*, *ICP27* and *gB* genes, were statistically decreased and the reduction of *L* gene *gB* was ascribed as the consequence of downregulation of *IE* genes *ICP4* and *ICP27*. This implied that the NPs were active in inhibiting HSV-2 replication. The suppression of HSV-2 replication might be caused by the cationic character of CS, enabling binding to negatively charged DNA or RNA [42], thereby interfering with the viral genome. Notably, more inhibitory effects were found when the infected cells were applied to the NPs containing EEP. Regarding these results, the active EEP was able to present intracellular activity based on the nano-delivery system, thereby increasing the therapeutic potential of EEP. EEP might interfere not only with the superficial structure of the virion as mentioned above, but with the bioactive substances derived from EEP such as flavonoids, also probably inhibiting viral polymerase, binding to the viral nucleic acid, thus interrupting nucleic acid synthesis [43,44]. This suggests dual effects resulting from CS and EEP. Similarly, another work also introduced the use of CS in nanosystems to increase antiviral activity when compared to the drug alone [23,45].

Since this is the first time that the encapsulation of EEP in CS-modified PLGA NPs was investigated for application in HSV-2 treatment, more investigations regarding its characteristics and molecular mechanisms of antiviral activity need to be carried out. We also suggest that this nanoformulation could be further designed as a new topical formulation to inhibit viral replication that exists in the epidermis or the basal layer.

## 4. Materials and Methods

### 4.1. Propolis, Chemicals, and Reagents

The ethanolic extract of propolis (EEP) was kindly provided by Bee Products Industry (Lamphun, Thailand). Poly(lactic-co-glycolic acid) (PLGA) (lactide:glycolide = 50:50; inherent viscosity = 0.45–0.60 dL/g, Mw = 38–54 kDa) was purchased from Sigma-Aldrich (St. Louis, MO, USA). Chitosan (CS) was kindly gifted by Assist. Prof. Dr. Worrapan Poomanee, Department of Pharmaceutical Sciences, Faculty of Pharmacy, Chiang Mai University. Polyvinyl alcohol (PVA) and ethanol (EtOH) were purchased from Fluka (Buchs, Switzerland) and Merck Millipore (Darmstadt, Germany), respectively. Dichloromethane (DCM) was obtained from RCI Labscan (Gliwice, Poland). All the other chemicals and reagents used in this study were of analytical and molecular grade.

### 4.2. Preparation and Characterization of EEP-Loaded CS/PLGA Nanoparticles (EEP-NPs)

#### 4.2.1. Preparation of EEP-NPs

Formulations of EEP-NPs (or Empty-NPs) were prepared using the modified oil-in-water (*o*/*w*) single emulsion solvent evaporation method with some modifications [9]. Briefly, an organic solution consisting of 100 mg/mL EEP and 100 mg PLGA dissolved in DCM was prepared and added dropwise to an aqueous solution containing 0.4% (*w*/*v*) PVA and 1% (*w*/*v*) CS to obtain the ratio of 1:2 (*v*/*v*) of organic and aqueous phases. The resulting solution was stirred, and then sonicated using an ultrasonic processor UP50H (Hielscher Ultrasonics, Hielscher, NJ, USA) at 90% amplitude for 30 min within an ice bath. For complete polymerization, each mixture was stored overnight at room temperature in the dark. Afterwards, the solution was centrifuged at 8800× *g* for 40 min at 4 °C to obtain NPs. The NPs were further washed once and reconstituted with deionized water before lyophilization. The NP samples were stored at −20 °C until used.

#### 4.2.2. Physicochemical Characterization

Particle size, polydispersity index (PDI), and zeta (ζ) potential (ZP)

The average particle size, polydispersity index (PDI) values, and zeta (ζ) potential (ZP) of the surface charge of the nanoformulations were evaluated using a Malvern Zetasizer Nano ZSP system (Malvern Instruments, Worcestershire, UK). The measurements were carried out for the EEP-NPs and the Empty-NPs at a 1/100 dilution in deionized water. Each sample was measured based on at least three measurements in three individual runs.

2.Scanning electron microscopy (SEM)

The morphology of the NPs was observed using a scanning electron microscope (SEM) (JEOL, Tokyo, Japan). A drop of the formulation was air-dried on a copper tape and coated with a gold film using a JEOL JFC1100E Ion Sputter (JEOL, Tokyo, Japan) under vacuum for 2 min. After preparation, the shape and size of the particles were studied using SEM at an accelerating voltage of 10 kV.

3.Encapsulation efficiency (EE) and loading capacity (LC)

The encapsulated EEP of the NPs was determined by means of an indirect quantification method. The supernatant of the EEP-NPs was obtained after centrifugation at 12,000× *g* for 1 h at 4 °C. The supernatant was properly diluted in deionized water, and the absorbance was then measured at 290 nm using a UV–Vis spectrophotometer. The amount of EEP in the supernatant was determined using the standard calibration curve of EEP. The EE and LC of the EEP-NPs were quantified using the following Equations (1) and (2):(1)$$\mathrm{EE}\;\left(\%\right)=\frac{\mathrm{amount}\;\mathrm{of}\;\mathrm{EEP}\;\mathrm{encapsulated}\;\mathrm{in}\;\mathrm{the}\;\mathrm{NPs}}{\mathrm{initial}\;\mathrm{EEP}\;\mathrm{added}}\times 100$$
(2)$$\mathrm{LC}\;\left(\%\right)=\frac{\mathrm{amount}\;\mathrm{of}\;\mathrm{EEP}\;\mathrm{in}\;\mathrm{the}\;\mathrm{NPs}}{\mathrm{total}\;\mathrm{amount}\;\mathrm{of}\;\mathrm{NPs}}\times 100$$

#### 4.2.3. In Vitro Release Assay

The EEP release from the NPs was performed using a modified dissolution method [40]. One milligram of the EEP-NPs was dissolved in 1 mL of a phosphate-buffered saline solution (PBS, pH 7.4 or 4.0) and shaken at 45 rpm in an incubator at 37 °C. The samples were collected at different timepoints (0, 1, 2, 3, 6, 12, 18, and 24 h) and centrifuged at 9800× *g* for 20 min. The supernatant was harvested, and the pellet was resuspended with 1 mL of fresh PBS. The release of EEP from the NPs in the supernatant at the specified time interval was measured using a UV–Vis spectrophotometer at 290 nm. The percentage of EEP released from the NPs (% accumulative release) was compared with the EEP’s standard calibration curve.

### 4.3. Cell Culture and Virus

African green monkey kidney epithelium (Vero) cells (ATCC CCL-81) were purchased from the American Type Culture Collection (ATCC, Manassas, VA, USA) and maintained in Dulbecco’s modified Eagle’s medium (DMEM) supplemented with 10% fetal bovine serum (FBS), penicillin (10 units/mL), and streptomycin (100 μg/mL) at 37 °C and 5% CO_2_. The HSV-2 virus was obtained from Prof. Dr. Chalobon Yoosook, Department of Microbiology, Faculty of Science, Mahidol University. Acyclovir (Sigma-Aldrich, Darmstadt, Germany) was kindly gifted by Assist. Prof. Dr. Yingmanee Tragoolpua, Department of Biology, Faculty of Sciences, Chiang Mai University.

### 4.4. Cytotoxicity of EEP-NPs

The effect of the EEP-NPs and the Empty-NPs on the viability of the Vero cells was determined by means of the trypan blue assay. The cells (1 × 10^4^ cells/well) were seeded in a 96-well plate. On the following day, different concentrations of the EEP-NPs (or the Empty-NPs) were added and incubated for 24 h. Then, the cells were harvested by trypsinization, and the number of live and dead cells was determined. The cytotoxicity of the NPs was expressed as the % cell viability, and the 50% cytotoxic concentration (CC_50_) was then calculated.

### 4.5. Assessment of Antiviral Activities

#### 4.5.1. Determination of the Virus Titer Using a Plaque Assay

A plaque assay was performed according to Deethae et al., 2018 [46], with some modifications. In brief, Vero cells (1 × 10^5^ cells/well) were seeded in a 24-well plate overnight. The cell monolayer was infected with the HSV-2 virus at various dilutions and incubated at 37 °C in a 5% CO_2_ incubator for 1 h. The infected cells were then overlaid with an overlay medium. After 48 h, the cells were washed three times with PBS and stained with 1% crystal violet solution. The plaques were counted, and plaque formation units (PFUs) were calculated.

#### 4.5.2. Investigation of Antiviral Activity of the EEP-NPs against HSV-2 Infection

For the treatment assay [46], the Vero cell monolayer was first incubated with 30 PFU/well of the HSV-2 virus at 37 °C for 1 h to allow viral attachment. Then, the EEP-NPs (or the Empty-NPs) with different concentrations were added to the infected cells, followed by an overlay medium. Quantification of the virus was performed using the plaque assay as described above (Section 4.5.1). The infected cells without treatment with the EEP-NPs (or the Empty-NPs) were used as the virus control. The antiviral activity was determined by the percentage of HSV plaque inhibition (%) using the following Equation (3):(3)$$\%\;\mathrm{Inhibition}=\frac{V_{C}-V_{T}}{V_{C}}\times 100$$
where V_C_ and V_T_ refer to the number of plaques in the absence and presence of EEP-NPs (or Empty-NPs).

Furthermore, the selectivity index (SI) was also calculated to indicate the toxicity of the EEP-NPs for normal cells as compared to viruses using the following Equation (4):(4)$$\mathrm{Selectivity}\;\mathrm{index}\;\left(\mathrm{SI}\right)=\frac{{\mathrm{CC}}_{50}}{{\mathrm{IC}}_{50}}\times 100$$
where CC50 and IC50 refer to the 50% cytotoxic concentrations and 50% inhibition concentrations, respectively.

#### 4.5.3. Direct Inactivation Assays

To examine whether the EEP-NPs directly inhibited the viral particles, a direct inactivation assay was performed according to Deethae et al., 2018, with some modifications [46]. Briefly, the HSV-2 virus supernatant at 200 PFU/mL was pretreated with the EEP-NPs (or the Empty-NPs) and kept at 4 °C for 0 min, 30 min, 6 h, and 24 h. Then, the treated viruses were added to the cells and incubated for another 48 h based on the plaque assay as described above (Section 4.5.1)

#### 4.5.4. Virus Release

The monolayer cell culture was infected with 30 PFU/well of the virus for 1 h to allow viral attachment. After 24 h treatment of the infected cells with the EEP-NPs (or the Empty-NPs), the supernatant and the cell pellet were collected, respectively, and the virus release was examined according to Wang et al., 2020 [40], with some modifications. The cell pellet was subjected to three freeze–thaw cycles before titration. The plaque assay was performed, and virus titers of the supernatant and the cell pellet were determined. The virus release rate after treatment with the EEP-NPs (or the Empty-NPs) was also calculated using the following Equation (5):(5)$$\mathrm{Virus}\;\mathrm{release}\;\mathrm{rate}\;\left(\%\right)=\frac{T_{\mathrm{ex}}}{T_{\mathrm{ex}}+T_{\mathrm{in}}}\times 100$$
where T_ex_ and T_in_ represent the extracellular and intracellular virus titers, respectively.

#### 4.5.5. Expression Levels of Immediate Early (IE) and Late (L) HSV-2 Genes Using Quantitative Real-Time PCR

For the analysis of mRNA expression of the immediate early *ICP4* and *ICP27* and glycoprotein (*gB*) HSV-2 genes, total RNA of the infected cells treated with the EEP-NPs (or the Empty-NPs) with a different post-infection (p.i.) time was extracted with the TRIzol reagent (Ambion, CA, USA) according to the manufacturer’s instructions and reverse-transcribed into cDNA using a Thermo Scientific RevertAid First Strand cDNA synthesis kit (Thermo Scientific, Waltham, MA, USA). The amplifications were carried out using a SensiFAST SYBR kit (BIOLINE, London, UK) using primers for *ICP4*, *ICP27,* and *gB*. Glyceraldehyde-3-phosphate dehydrogenase (*GAPDH*) was used as the internal control. The sequence of primers used in the PCR analysis is shown in Table 3. The PCR reactions included initial denaturation at 95 °C for 120 s, followed by 40 cycles of denaturation at 95 °C for 10 s and annealing at 59 °C (*ICP4* and *ICP27*) or 61 °C (*gB*) for 31 s. The relative fold change in mRNA expression was analyzed using the 2^−^^ΔΔCT^ method. All the experiments were performed in three independent trials.

### 4.6. Fluorescence Staining Assay

The localization of the EEP-NPs was investigated using a confocal laser scanning microscope (CLSM). The cell monolayer was infected with HSV-2 and incubated at 37 °C, 5% CO_2_, for 1 h. To track the EEP-NPs, a 500 μL sample of the EEP-NPs was stained with 40 μL of 0.25 mg/mL Nile red (NR) solution for 20 min in the dark, at room temperature. After washing, the pellet was resuspended and subjected to treatment with the infected cells for 24 h. Then, the cells were fixed with 4% paraformaldehyde for 20 min, followed by a wash with PBS. Afterwards, the fixed cells were blocked with 2% bovine serum albumin at room temperature for 30 min and incubated with a mouse anti-α_v_-integrin mAb (kindly gifted by Prof. André Lieber, Department of Medicine, University of Washington, Seattle, WA, USA). After being washed, the cells were incubated with an FITC-conjugated rabbit anti-mouse polyclonal antibody (Dako, Santa Clara, CA, USA) (kindly gifted by Assoc. Prof. Dr. Sawitree Chiampanichayakul, Department of Medical Technology, Faculty of Associated Medical Sciences, Chiang Mai University). The cell nuclei were then counterstained with 4′,6-diamidio-2-phenylindole (DAPI) (Invitrogen, Thermo Fisher Scientific, Waltham, MA, USA). Images were acquired using a confocal laser scanning microscope (Leica TCS SP8, Mannheim, Germany).

### 4.7. Statistical Analysis

The results are expressed as the means ± SEM. All the experiments were repeated in three independent trials. Comparisons between the groups were performed using a one-way analysis of variance and post hoc LSD. Significant differences were indicated by *p* < 0.05 or 0.001. All the calculations were performed using the SPSS Statistics 23.0 software (IBM Corp., Armonk, NY, USA).

## 5. Conclusions

In this work, the improvement of characteristics regarding the surface charge of the PLGA-based NPs loaded with EEP was achieved by CS modification. The EEP-NPs demonstrated the ability to inactivate HSV-2 particles and block the steps of viral entry and release. Additionally, the EEP-NPs exhibited molecular biological activity on the HSV-2 replication process through downregulation at the transcriptional level of *ICP4*, *ICP27,* and *gB*. According to these results, the use of EEP based on a nano-drug delivery system can provide utility in various aspects that could be further developed as an alternative platform to treat HSV-2 infections.

## Figures and Tables

**Figure 1 molecules-27-02560-f001:**
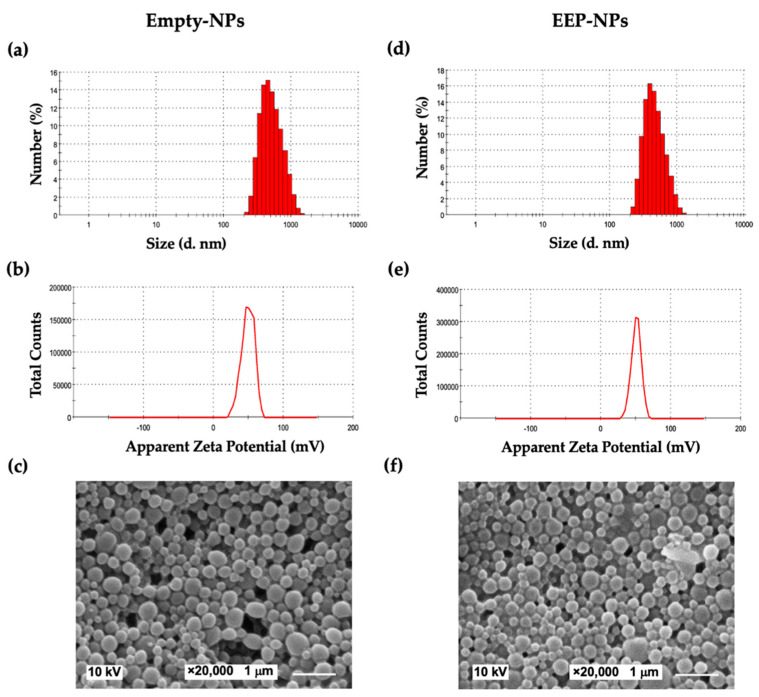
Characteristics of the NPs. Morphology images, size distribution and ζ potential (ZP) curves of the Empty-NPs (**a**–**c**) and the EEP-NPs (**d**–**f**). The scale bar represents 1 μm.

**Figure 2 molecules-27-02560-f002:**
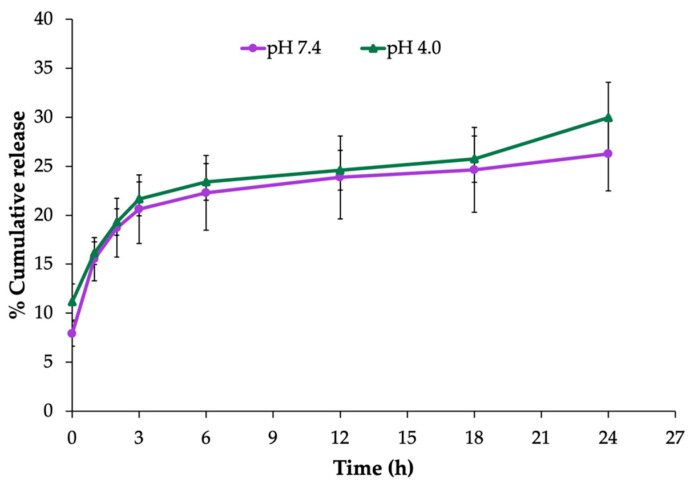
In vitro release profile of the EEP-NPs in PBS, pH 7.4 and pH 4.0, at 37 °C for 24 h. The data are represented as the means ± SEM of three independent trials.

**Figure 3 molecules-27-02560-f003:**
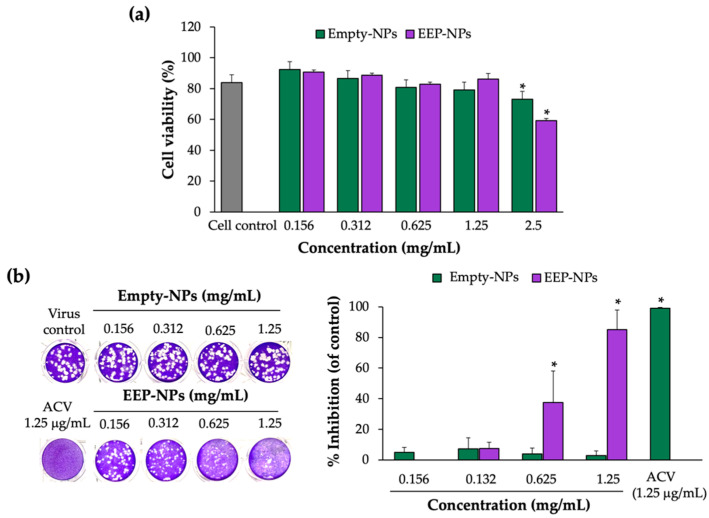
Cytotoxicity and anti-HSV-2 activity of the EEP-NPs. (**a**) Determination of cytotoxicity of the NPs on the Vero cells. The cells were treated with various concentrations of the EEP-NPs (or the Empty-NPs), and cell viability was evaluated by means of the trypan blue exclusion assay. (**b**) Dose–response effect of the EEP-NPs against HSV-2. The HSV-2-infected cells were treated with different concentrations of the EEP-NPs, Empty-NPs, or ACV at IC_90_ (1.25 μg/mL). The antiviral activity was determined by means of the plaque reduction assay. The results represent the means ± SEM from three independent experiments performed in triplicate. Note: * *p* < 0.05—significant compared to the virus control.

**Figure 4 molecules-27-02560-f004:**
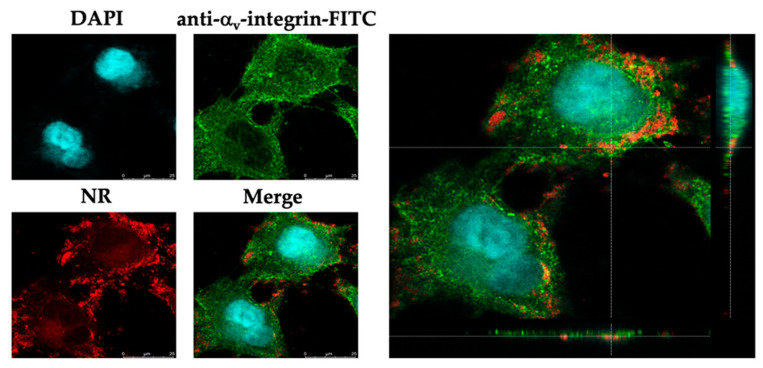
Localization of the EEP-NP-treated cells. XY and orthogonal images demonstrating cellular uptake of the EEP-NPs in the HSV-2-infected Vero cells. The EEP-NPs were traced with NR (red). The nucleus and the cell membrane were indicated using DAPI staining (blue) and a mouse anti-α_v_-integrin mAb followed by an FITC-conjugated anti-mouse pAb (green), respectively.

**Figure 5 molecules-27-02560-f005:**
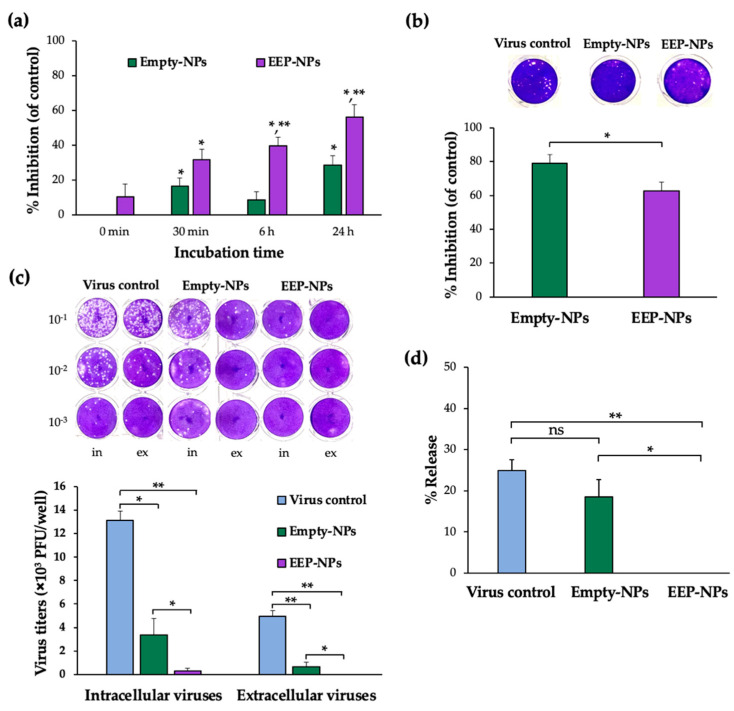
Effect of the EEP-NPs on HSV-2 activity in different modes. (**a**) The direct inactivation effect of the EEP-NPs against viral particles was evaluated by means of the plaque formation assay. The viral particles were pretreated with the EEP-NPs (or the Empty-NPs) at 4 °C for 30 min, 6 h, and 24 h. Note: * *p* < 0.05—significant compared to the virus control; ** *p* < 0.05—significant compared to the Empty-NPs. (**b**) The inhibition of viral entry was investigated after the cells were preincubated with the EEP-NPs (or the Empty-NPs) for 24 h. The reduction of plaque formation was measured and represented as a percentage of inhibition of viral entry. Note: * *p* < 0.05—significant compared to the Empty-NPs. (**c**) The inhibitory effect of the EEP-NPs on intracellular and extracellular viruses. The infected cells were treated with the EEP-NPs (or the Empty-NPs). After 24 h of treatment, the virus titers of the infected cells and the culture supernatant were determined by means of the plaque titration assay, and (**d**) the virus release rate percentage was determined. Note: * *p* < 0.05 and ** *p* < 0.001.

**Figure 6 molecules-27-02560-f006:**
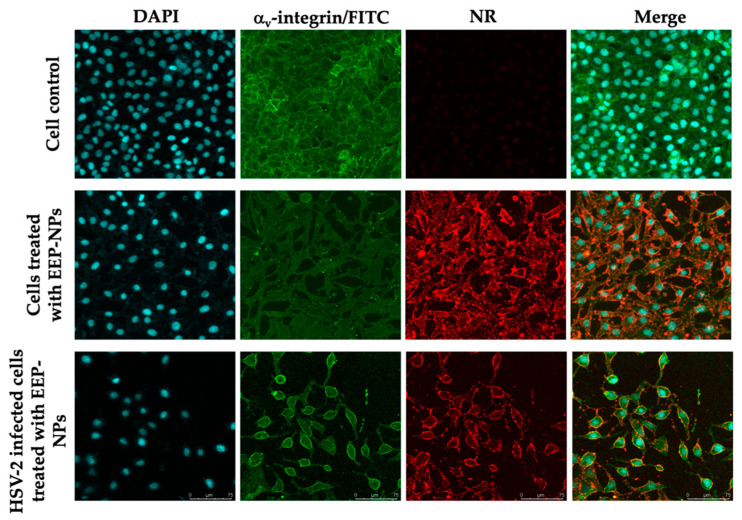
Blocking of virus entry and release by the EEP-NPs. The EEP-NPs were traced with NR (red). The nucleus and the membrane were indicated using DAPI staining (blue) and a mouse anti-α_v_-integrin mAb followed by an FITC-conjugated anti-mouse pAb (green), respectively.

**Figure 7 molecules-27-02560-f007:**
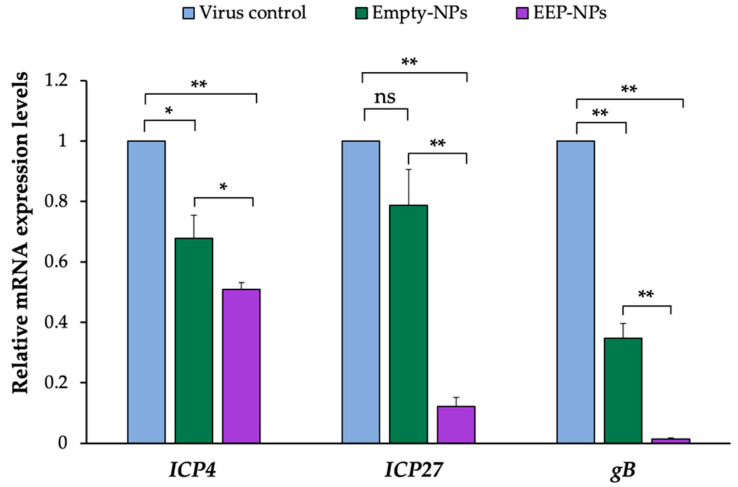
Effect of the EEP-NPs on the HSV-2 IE and L gene expression. The infected cells were treated with the EEP-NPs (or the Empty-NPs). The mRNA expression levels of *ICP4* and *ICP27* (4 h p.i.) and *gB* (24 h p.i.) were assessed by means of quantitative real-time PCR. Relative mRNA expression was normalized to *GAPDH* as the internal control, and the expression levels were represented as a fold change of 2^−ΔΔCT^ compared to the virus control. Note: * *p* < 0.05, ** *p* < 0.001.

**Table 1 molecules-27-02560-t001:** Particle size, polydispersity index (PDI), ζ potential (ZP), and entrapment efficiency (EE) of the NPs.

Parameters	Empty-NPs	EEP-NPs
Size (nm)	651.35 ± 59.19	447.15 ± 31.66
PDI	0.34 ± 0.16	0.21 ± 0.06
ZP (mV)	36.75 ± 8.17	38.05 ± 7.65
EE (%)	ND	79.89 ± 13.92
LC (%)	ND	37.68 ± 6.56

**Table 2 molecules-27-02560-t002:** Cytotoxicity concentrations (CC), inhibition concentrations (IC), and selectivity index (SI) of the NPs.

Formulations	CC_50_ (mg/mL ± SEM)	IC_50_ (mg/mL ± SEM)	SI (CC_50_/IC_50_)
Empty-NPs	>2.5	>1.25	-
EEP-NPs	>2.5	0.80 ± 0.16	>3.12

CC_50_: 50% cytotoxic concentrations; IC_50_: 50% inhibition concentrations.

**Table 3 molecules-27-02560-t003:** Sequence of primers used in the real-time PCR amplifications.

Primers	Sequences (5′-3′)	Ref.
*ICP4*	GATGGGGTGGCTCCAGAAC	[40]
	AGATGAAGGAGCTGCTGTTGC	
*ICP27*	CCCTTTCTGCAGTGCTACCT	[40]
	CCTTAATGTCCGACAGGCGT	
*gB*	CCATGACCAAGTGGCAGGAG	[40]
	AGGTTGGTGGTGAAGGTGGTC	
*GAPDH*	GAAGGTGAAGGTCGGAGTC	[47]
	GAAGATGGTGATGGGATTTC	

## Data Availability

Not applicable.

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
