# Peer review of "Activity of Propolis Nanoparticles against HSV-2: Promising Approach to Inhibiting Infection and Replication"

_molecules, 2022, doi:10.3390/molecules27082560_

Round 1

Reviewer 1 Report

In my opinion, the study presents interesting results on the activity of propolis nanoparticles against HSV-2. However, there are some comments that need to be addressed and clarified before further consideration.

  1. Section (4.5. Assessment of antiviral activities) and its subsections. Please provide references to all used methods. This is to certify that you have used proper methods for determining the antiviral activity.
  2. Regarding statistical analyses. Please provide the post-hoc comparison tests that were used to assess the differences between treatments with test compounds and positive control. As shown in your results that you have used a one-way ANOVA test. This is not enough to confirm the differences. 
  3. Why did not the author calculate the selectivity index for EEP-NPs? Please provide this information. 
  4. Finally, I recommend that the authors double-check the full text for grammatical and typing errors. 

Author Response

Response to Reviewer 1:

  1. Section (4.5. Assessment of antiviral activities) and its subsections. Please provide references to all used methods. This is to certify that you have used proper methods for determining the antiviral activity.

Response: As reviewer’s suggestion, we have added the references and more detail of the methods for antiviral assessment in the revised manuscript at page 13, line 448-449, line 456-457, 478-481, and 486-489 as shown below.

4.5. Assessment of antiviral activities

4.5.1. Determination of virus titer by plaque assay

Plaque assay was performed according to Deethae et al., 2018 [1] with some modification. In brief, the Vero cells (1x105 cells/well) were seeded in a 24-well plate overnight. The cell monolayer was infected with HSV-2 virus at various dilutions and incubated at 37°C, 5% CO2 incubator for 1 h. The infected cells were then overlaid with an overlay medium. After 48 h, the cells were washed three times with PBS and stained with 1% crystal violet solution. The plaques were counted, and plaque formation units (PFUs) were calculated.

4.5.2. Investigation of antiviral activity of EEP-NPs against HSV-2 infection

For treatment assay [1], the Vero cell monolayer was firstly incubated with 30 PFU/well of HSV-2 virus at 37°C for 1 h to allow viral attachment. Then, the EEP-NPs (or Empty-NPs) with different concentrations were added to the infected cells, followed by an overlay medium. Quantification of the virus was performed by plaque assay as described above (section 4.5.1). The infected cells without treatment of EEP-NPs (or Empty-NPs) were used as a virus control. The antiviral activity was determined by the percentage of HSV plaque inhibition (%) using the following equation (3):

% Inhibition = (VC – VT) ´ 100                                                                     (3)

                             VC

Where VC and VT refer to the plaque’s number in the absence and presence of EEP-NPs (or Empty-NPs).

Furthermore, the selectivity index (SI) was also calculated to indicate the toxicity of EEP-NPs to normal cells as compared to the viruses using the following equation (4):

    Selectivity index (SI) = CC50                                                         (4)

                                         IC50

     Where CC50 and IC50 refer to the 50% cytotoxic concentrations and 50% inhibition concentrations, respectively.

4.5.3. Direct inactivation assays

To examine whether EEP-NPs directly inhibited the viral particles, direct inactivation assay was performed according to Deethae et al., 2018 with some modification [1]. Briefly, the HSV-2 virus supernatant at 200 PFU/mL was pre-treated with EEP-NPs (or Empty-NPs) and kept at 4°C for 0 min, 30 min, 6 h and 24 h. Then, the treated viruses were added to the cells and incubated for another 48-h based on the plaque assay as described above (section 4.5.1)

4.5.4. Virus release

The monolayer cell culture was infected with 30 PFU/well of the virus for 1 h to allow viral attachment. After 24 h treatment of the infected cells with EEP-NPs (or Empty-NPs), the supernatant and cell pellet were collected, respectively and the virus release was examined according to Wang et al., 2020 [2] with some modification. The cell-pellet was subjected to three freeze-thaw cycles before titration. The plaque assay was performed, and virus titers of supernatant and cell pellet were determined. The virus release rate after treatment with EEP-NPs (or Empty-NPs) was also calculated using the following equation (5):

    Virus release rate (%) =        Tex       x 100                                (5)

                                           (Tex + Tin)

      Where Tex and Tin represent extracellular and intracellular virus titer, respectively.

  1. Regarding statistical analyses. Please provide the post-hoc comparison tests that were used to assess the differences between treatments with test compounds and positive control. As shown in your results that you have used a one-way ANOVA test. This is not enough to confirm the differences. 

Response: The different results between groups were analyzed by One-Way ANOVA and LSD post-hoc comparison. As reviewer’s suggestion, we have added more detail of the comparison in section 4.7 of the revised manuscript at page 15, line 539-540 as followed “Comparisons between groups were performed using a one-way analysis of variance and LSD post-hoc.”  

  1. Why did not the author calculate the selectivity index for EEP-NPs? Please provide this information. 

Response: As reviewer’s suggestion, we have provided the value of selectivity index (SI) of EEP-NPs and added the information of the 50% cytotoxicity concentrations (CC50), 50% inhibition concentrations (IC50) and SI in the revised manuscript as shown below.

In page 4, line 136-137and 141-145

To investigate the anti-HSV-2 activity of EEP-NPs, infected cells were treated with various concentrations of NPs for 24 h and the reduction of plaque formation was used to evaluate the anti-HSV-2 activity by EEP-NPs. As shown in Figure 3b and Table 2, EEP-NPs inhibited the plaque formation and showed a significant percentage of plaque inhibition in a dose dependent manner with the 50% inhibition concentration (IC50) at 0.80±0.16 mg/mL. Meanwhile, Empty-NPs had no effect. The selectivity index (SI) of EEP-NPs was more than 3.12, indicating non-toxic agent and further studies can be performed. In addition, the administration of EEP-NPs at 1.25 mg/mL showed comparable activity to ACV (IC90). Thus, the highest concentration of NPs (1.25 mg/mL) was used to investigate the mode of action in further experiments.

In page 10, line 296-299

Subsequently, the low toxic concentrations of EEP-NPs were tested for antiviral activity. Our results showed the antiviral effects of EEP-NPs after treatment of the HSV-2 infected cells for 24 h in a dose-dependent manner. Moreover, the SI value also indicated non-toxic of EEP-NPs as previously described [3]. However, many variations of acceptance criteria of SI values have been reported. Thus, further investigation is still required. Then, the CLSM analysis was conducted to show the uptake of NPs into the infected cells (Figure 4). The result demonstrated that the EEP-NPs could enter the infected cells and were located mainly in cell membrane, following by cytoplasm.

In page 13, line 444-445

4.4. Cytotoxicity of EEP-NPs

The effect of EEP-NPs and Empty-NPs on the viability of Vero cells was determined by trypan blue assay. The cells (1´104 cells/well) were seeded in a 96-well plate. On the following day, different concentrations of EEP-NPs (or Empty-NPs) were added and incubated for 24 h. Then, the cells were harvested by trypsinization, and the number of live and dead cells was determined. The cytotoxicity of NPs was expressed as the % cell viability and the 50% cytotoxic concentration (CC50) was then calculated.

In page 13, line 469-476

Furthermore, the selectivity index (SI) was also calculated to indicate the toxicity of EEP-NPs to normal cells as compared to the viruses using the following equation (4):

Selectivity index (SI) = CC50                                                                (4)

                                    IC50

     Where CC50 and IC50 refer to the 50% cytotoxic concentrations and 50% inhibition concentrations, respectively.

Moreover, we also provided the Table representing the CC50, IC50 and SI values of NPs in the revised manuscript at page 5, line 160-164 as shown below.

Table 2. Cytotoxicity concentrations (CC), inhibition concentrations (IC), and selectivity index (SI) of NPs.

Formulations

CC50

(mg/mL ± SEM)

IC50

(mg/mL ± SEM)

SI

(CC50/IC50)

Empty-NPs

> 2.5

> 1.25

-

EEP-NPs

> 2.5

0.80±0.16

> 3.12

CC50: 50% cytotoxic concentrations; IC50: 50% inhibition concentrations

  1. Finally, I recommend that the authors double-check the full text for grammatical and typing errors. 

Response: We have re-checked and corrected typing errors in the revised manuscript according to reviewer’s suggestion.

References

  1. Deethae, A., et al., Inhibitory effect of Spirogyra spp. algal extracts against herpes simplex virus type 1 and 2 infection. J Appl Microbiol, 2018. 124(6): p. 1441-1453.
  2. Wang, L., et al., Antiviral mechanism of carvacrol on HSV-2 infectivity through inhibition of RIP3-mediated programmed cell necrosis pathway and ubiquitin-proteasome system in BSC-1 cells. BMC Infect Dis, 2020. 20(1): p. 832.
  3. Indrayanto, G., G.S. Putra, and F. Suhud, Validation of in-vitro bioassay methods: Application in herbal drug research. Profiles Drug Subst Excip Relat Methodol, 2021. 46: p. 273-307.

Reviewer 2 Report

Comments:

I reviewed the manuscript entitled, ‘Activity of Propolis Nanoparticles Against HSV-2: promising approach for inhibiting infection and replication’. The manuscript is well written, authors must consider the following points.

Minor comments:

Line 28 :  “single-emulsion solvent evaporation method” rewrite as “single-emulsion by using solvent evaporation method” or revise the sentence.

Line 29 – 31: Explain the results with average mean and standard deviation (values).

Line 96: our previous work…..Cite reference

Line 100 – 101: correct Pdi to PDI and throughout the manuscript

Line 102: Expand LC values

Line: 185: significantly between EEP-NPs and Empty-NPs……………..significantly lower or higher?

Line 387 and 493: Correct 2 mins…will be 2 min

Major comments:

Line 99 – 101: Why empty NPs have less size as compared to the encapsulated NPs?

Fig 1: SEM images are not clear, with magnification? It would be better to show the images separately with magnification.

Fig 5 a : Empty NPs showed % inhibition?....... justify

Fig 5 d : Virus release % was zero in case EEP-NPs ? how much it reduced the viral count…….what was the initial count cell for viral cells?

Line 183 – 184: EEP-NPs and Empty-NPs showed a remarkable plaque inhibition, approximately 60% and 80%, respectively…..why?

Author Response

Minor comments:

  1. Line 28: “single-emulsion solvent evaporation method” rewrite as “single-emulsion by using solvent evaporation method” or revise the sentence.

    Line 29 – 31: Explain the results with average mean and standard deviation (values).

Response: As reviewer’s suggestion, we have re-written and extended the results in the abstract of the revised manuscript at page 1, line 26-31 as followed “An ethanolic extract of propolis (EEP) was encapsulated in nanoparticles composed of poly lactic-co-glycolic acid and chitosan, using the modified oil-in-water single-emulsion by using solvent evaporation method. The produced nanoparticles (EEP-NPs) had a spherical shape with a size of ~450 nm and presented the satisfied physicochemical properties including positively charged surface (38.05±7.65 mV), highly entrapment efficiency (79.89±13.92%), and sustained release-profile.”

  1. Line 96: our previous work…Cite reference

Response: We have added the reference of this work in the revised manuscript at page 2, line 96-97.

  1. Line 100 – 101: correct Pdi to PDI and throughout the manuscript

Response: We have changed the word “PdI” to “PDI” in the revised manuscript at page 3, line 100-101 and Table 1; page 9, line 282 and page 12, line 394.

  1. Line 102: Expand LC values

Response: The word “LC values” is the value of loading capacity (LC). As reviewer’s suggestion, we have expanded this word in the title of section 2.1.1 of the revised manuscript at page 2, line 95 as followed “2.1.1. Particle Size, Polydispersity Index (PDI), Zeta Potential (ZP), Entrapment Efficiency (EE), Loading Capacity (LC), and Morphology”.

  1. Line: 185: significantly between EEP-NPs and Empty-NPs…………significantly lower or higher?

Response: In the study of effect of EEP-NPs on virus entry, the percentage of plaque inhibition was significant lower after EEP-NPs treatment when compared to Empty-NPs. As reviewer mentioned, we have added more description of this result in the revised manuscript at page 6, line 195-196 as followed “Moreover, the percentage of inhibition was significant lower in EEP-NPs when compared to Empty-NPs.”

  1. Line 387 and 493: Correct 2 mins…will be 2 min

Response: We have corrected the word “mins” to “min” in the revised manuscript at page 12, line 404 and page 14, line 526. 

Major comments: 

  1. Line 99 – 101: Why empty NPs have less size as compared to the encapsulated NPs?

Response: From this study, the EEP-NPs showed a slightly smaller size than Empty-NPs. This result might be due to the effect of EEP on packing and distribution of PLGA and CS in the formulation. Incorporation of EEP solution in the hydrophilic layer might cause the structure of particle is more packed than empty particle. This phenomenon was concordant with previous work described by Obeid et al., 2020 [1].  

  1. Fig 1: SEM images are not clear, with magnification? It would be better to show the images separately with magnification.

Response: As reviewer’s suggestion, we have separated the SEM images with magnification and edited Figure numbers in the revised manuscript at page 3, as shown below.

Figure 1. Characteristics of NPs. Morphology images, size distribution and ζ potential (ZP) curves of Empty-NPs (a, b, and c) and EEP-NPs (d, e, and f). The scale bar represents 1 µm.

  1. Fig 5a: Empty NPs showed % inhibition?....... justify

Response: From the result shown in Figure 5a, Empty-NPs also had an effect on plaque inhibition, influencing by the intrinsic antiviral activity of CS as mentioned in the discussion of the revised manuscript page 10, line 305-306 and 322-329, describing that “Empty-NPs also inactivated HSV-2 virions possibly due to the degradation of NPs and the release of cationic CS. CS has an intrinsic antimicrobial property including antiviral activity [2]. It has been proposed that the electrostatic interaction of positively charged CS and the negatively charged surface of the virus can inhibit the viral activity and/or directly kill the virus through viral protective membrane disruption [2, 3]. Moreover, the viral capsid proteins or other virus-specific proteins could be interrupted by CS or its derivatives, leading to the prevention of viral glycoproteins and the interaction with their receptors, the reduction of viral entry, and eventually the suppression of viral replication [2, 4, 5].”

  1. Fig 5d: Virus release % was zero in case EEP-NPs? how much it reduced the viral count…what was the initial count cell for viral cells?

Response: For the study on antiviral activities of EEP-NPs, the initial cell number was 1´105 cells/well and the monolayer cells were then infected with the virus at 30 PFU/well. After 1 h viral adsorption, the medium was removed, and the infected cells were then treated with EEP-NPs for 24 h. From the result in Figure 5d, the virus release (%) in the culture supernatant was zero after treatment with EEP-NPs for 24 h, concordant to the results of plaque number and virus titers as shown in Figure 5c that was not found the extracellular virus. Meanwhile, some viral particles remained inside the cells or intracellular viruses.  

  1. Line 183 – 184: EEP-NPs and Empty-NPs showed a remarkable plaque inhibition, approximately 60% and 80%, respectively…why?

Response: For the study on virus entry, the cells were pre-incubated with EEP-NPs (or Empty-NPs) for 24 h before exposure to the virus. The result of plaque inhibition was 60% and 80% in EEP-NPs and Empty-NPs treatments, respectively. More activity of Empty-NPs on plaque formation might be a result of CS. As the positively charged CS can interact with the negatively charged cell membrane via the ionic adsorption reaction. Therefore, the CS contents in Empty-NPs can dissociate, attach the cell surface, and block viral entry. While incorporation of EEP in the formulation might be interfere or neutralize the surface charge of CS. Thus, blocking of viral entry to the cells by EEP-NPs show less activity than Empty-NPs.

According to this comment, we have added more explanations in the discussion of the revised manuscript at page 10, line 309-314 as followed “ Likewise, pre-incubation of the cells with the cationic NPs (EEP-NPs or Empty-NPs) were gathered with the negatively charged cell surface [6], ultimately viral attachment, adsorption, and entry processes were blocked. Noticeably, Empty-NPs showed more activity than that treatment with EEP-NPs in this step. This effect was probably influenced by dissociation of CS from the Empty-NPs, leading to an ionic adsorption between CS and cell surface. While incorporation of EEP in the formulation might result in interfering or neutralizing the surface charge of CS, therefore, accumulation of CS on the cell surface may be reduced.”  

References

  1. Obeid, M.A., et al., Comparison of the physical characteristics of monodisperse non-ionic surfactant vesicles (NISV) prepared using different manufacturing methods. Int J Pharm, 2017. 521(1-2): p. 54-60.
  2. Jaber, N., et al., A review of the antiviral activity of Chitosan, including patented applications and its potential use against COVID-19. J Appl Microbiol, 2022. 132(1): p. 41-58.
  3. Raafat, D. and H.G. Sahl, Chitosan and its antimicrobial potential--a critical literature survey. Microb Biotechnol, 2009. 2(2): p. 186-201.
  4. Boroumand, H., et al., Chitosan-Based Nanoparticles Against Viral Infections. Front Cell Infect Microbiol, 2021. 11: p. 643953.
  5. Gao, Y., et al., The inhibitory effects and mechanisms of 3,6-O-sulfated chitosan against human papillomavirus infection. Carbohydr Polym, 2018. 198: p. 329-338.
  6. Lu, B., X. Lv, and Y. Le, Chitosan-Modified PLGA Nanoparticles for Control-Released Drug Delivery. Polymers (Basel), 2019. 11(2).

Round 2

Reviewer 1 Report

The manuscript has been significantly improved. 

Reviewer 2 Report

All the comments have been addressed point-to-point by the authors, so I recommend the publication of this paper in Molecules.